# PQBP1: The Key to Intellectual Disability, Neurodegenerative Diseases, and Innate Immunity

**DOI:** 10.3390/ijms23116227

**Published:** 2022-06-02

**Authors:** Hikari Tanaka, Hitoshi Okazawa

**Affiliations:** Department of Neuropathology, Tokyo Medical and Dental University, 1-5-45 Yushima, Bunkyo-ku, Tokyo 113-8510, Japan; tanaka.npat@mri.tmd.ac.jp

**Keywords:** PQBP1, intellectual disability, neurodegenerative diseases, innate immunity, intrinsically disordered protein

## Abstract

The idea that a common pathology underlies various neurodegenerative diseases and dementias has attracted considerable attention in the basic and medical sciences. Polyglutamine binding protein-1 (PQBP1) was identified in 1998 after a molecule was predicted to bind to polyglutamine tract amino acid sequences, which are associated with a family of neurodegenerative disorders called polyglutamine diseases. Hereditary gene mutations of *PQBP1* cause intellectual disability, whereas acquired loss of function of PQBP1 contributes to dementia pathology. PQBP1 functions in innate immune cells as an intracellular receptor that recognizes pathogens and neurodegenerative proteins. It is an intrinsically disordered protein that generates intracellular foci, similar to other neurodegenerative disease proteins such as TDP43, FUS, and hnRNPs. The knowledge accumulated over more than 20 years has given rise to a new concept that shifts in the equilibrium between physiological and pathological processes have their basis in the dysregulation of common protein structure-linked molecular mechanisms.

## 1. Introduction

### 1.1. History of the Concept of Polyglutamine Disease

Albert La Spada, Kenneth H Fischbeck, and their colleagues were the first to develop the concept of polyglutamine (polyQ) disease by discovery of the causative gene for X-linked spinobulbar muscular atrophy (SBMA)/Kennedy’s disease [1]. The identified gene was an androgen receptor, in which the CAG repeat expansion was linked to patients but not healthy siblings in the same family. The CAG repeat was located in the exon and transcribed into a mutant protein with polyQ repeats. Thereafter, multiple causative genes of neurological and neuromuscular diseases–which included Huntington’s disease [2] and spinocerebellar ataxia [3]–that involve the expansion of triplet repeats were discovered, either in translated exons or untranslated regions, including introns. To date, more than 40 triplet diseases have been identified. Triplet diseases caused by repeat expansions in untranslated regions include Fragile X syndrome, which is caused by an elongation of a CGG repeat in the 5′ untranslated region of the FMR1 gene on the X chromosome, and myotonic dystrophy, which is caused by a CTG repeat in the untranslated region. Triplet diseases caused by repeat expansions in exons include Huntington’s disease, spinocerebellar ataxia types 1, 2, 3, 6, 7, and 17 (SCA1, 2, 3, 6, 7, and 17), dentatorubropallidoluysian atrophy (DRPLA), and Kennedy’s disease/SBMA, which are called polyQ diseases because abnormal proteins–including those with polyQ repeats–are thought to be linked to toxicity [4,5,6,7]. Many studies have supported this hypothesis by demonstrating that abnormal polyQ proteins induce toxicity via the formation of nuclear/cytoplasmic aggregates or inclusions [8,9,10]. Furthermore, X-ray diffraction revealed that the polyQ sequence, which adopts a β-sheet structure, functions as a polar zipper to generate aggregates [11].

The concept of polyQ disease has raised further questions regarding the mechanisms of toxicity. The first idea was “sequestration”, whereby normal proteins are co-segregated into the inclusion body of polyQ proteins and lose their physiological functions [12]. The second idea was that mutant polyQ proteins acquire higher (or lower) affinity for their binding partner molecules [13], and such gains in abnormal interactions (or loss of physiological interactions) lead to gains in toxicity.

In fact, some polyQ disease proteins interact with target proteins via sequences other than the polyQ tract [14,15]. Insertion of the polyQ sequence into an unrelated gene (e.g., the hypoxanthine phosphoribosyltransferase gene) induces neurodegeneration and inclusion body formation in knock-in mice [16]. However, it remains unclear whether the functional depletion of the protein due to abnormal aggregation or the gain of abnormal interactions with a new target protein due to inserted polyQ sequence caused neurodegeneration. Analyses using yeast cells have revealed that flanking sequences can have an effect on the toxicity of polyQ peptides [17], although it is uncertain whether the aggregation or the binding to a polyQ-interacting target is affected. Nevertheless, huntingtin (Htt)-exon 1 transgenic mice expressing an elongated CAG repeat peptide flanked with short sequences induce neurodegeneration [18], and the expression of polyQ peptides fused to fluorescent proteins is toxic in nematodes in a polyQ length-dependent manner [19,20], which suggests direct toxicity of the polyQ sequence. However, it remains unclear overall whether the change of aggregation or that of interaction causes the pathology.

### 1.2. History of the Discovery of PQBP1

PolyQ sequences are present in several transcription-related proteins, such as the transcription initiating factor TFIID/TATA-binding protein, glucocorticoid receptors, octamer-binding POU transcription factors, and the CREB-binding protein. Moreover, it is predicted that a certain cofactor binds to the polyQ sequence of Oct-2 because the polyQ sequence functions as an activation domain [21].

Therefore, our group hypothesized that certain molecules that interact directly with the polyQ repeat sequence mediate toxicity via their functional depletions and that such molecules are involved in the common pathogenesis of various polyQ diseases. To verify this hypothesis, we performed two-hybrid screening using a human embryonic brain cDNA library constructed on the pJG4–5 plasmid with a normal-length polyQ sequence (GAL-Q26-APP) derived from brain-specific transcription factor Brn2 as bait [22]. The proteins discovered were named PQBP1, PQBP2, PQBP3, PQBP4, and PQBP5 [20] and possessed polar amino acid-rich sequences, but no common structure or binding motif for interaction with the polyQ sequence [22].

Among these PQBPs, PQBP1 has been most extensively investigated. The cellular and biological functions of PQBP1 have been elucidated according to two specific domain structures of PQBP1, and various animal models of PQBP1 have been developed and investigated. Mutations of the human PQBP1 gene were shown to cause intellectual disability and other relevant symptoms. In addition, PQBP1 has been implicated in neurodegenerative diseases, including Alzheimer’s disease (AD) [23], tauopathy [24], and polyQ diseases [25], as a common mediator across multiple disease pathologies. In this review, we introduce the most recent overview of such accumulated knowledge of PQBP1, and present future prospects.

## 2. Molecular Structure and Cellular Function of PQBP1

PQBP1 has a WW domain (WWD) that is homologous to the SH3 domain in local protein structure and target sequence recognition [26,27,28,29,30,31,32], and a specific C-terminal domain (CTD) that is highly degenerated and classified as a low complexity domain/region [33] or an intrinsically disordered protein [34,35,36]. PQBP1-WWD is conserved from *C. elegans* to mammals and recognizes a short proline-rich sequence in various target proteins [27,31,37], which include some of the repeat sequences in the C-terminal tail of RNA polymerase II (Pol II) [25]. Because the interaction with Pol II occurs in a phosphorylation-dependent manner, PQBP1 preferentially binds to active Pol II with a phosphorylated tail, which transcribes and elongates pre-messenger RNA [25].

PQBP1-WWD is used for interaction with WWD-binding protein 11 (WBP11)/Npw38-binding protein (NpwBP). The interaction with WBP11 may be relevant to RNA splicing because mass analysis of different stages of spliceosomes assembly has revealed that PQBP1 and WBP11 are simultaneously incorporated into the B complex and released from the B * complex [38,39].

The CTD is unique to PQBP1 yet conserved across species in PQBP1 homologs [40]. Molecular structure analysis of the PQBP1-CTD revealed that it is highly disordered [41]. Circular dichroism (CD), nuclear magnetic resonance (NMR), and ensemble optimization method (EOM) analyses also confirmed that PQBP-CTD is an intrinsically disordered domain [42]. Therefore, PQBP1 is an intrinsically disordered protein (IDP). PQBP1 self-assembles via the intrinsically disordered structure of the CTD whilst it can revert to a monomer [42].

The target of PQBP1-CTD was identified as U5-15kD, a component of the U5 spliceosome, using yeast two-hybrid screening [43]. In an independent study of yeast, PQBP1 was also identified as a binding protein of Dim1p–a homolog of U5-15kD [44]–which further confirmed the relationship. The interaction of PQBP1 with the splicing protein is wellcoordinated with transcription. PQBP1 is recruited to the C-terminal tail of Pol II during active transcription, which is consistent with the observation that transcription of pre-mRNA is directly coupled with RNA splicing by hnRNPs and spliceosomes on the C-terminal tail of Pol II. A recent structural biology study revealed that the YxxPxxVL motif in PQBP1-CTD is essential for its interaction with U5-15kD [41,45]. Interestingly, the interaction motif is lost in all frameshift type mutations in a human intellectual disability called Renpenning syndrome [45], as described later in this review.

Hepta and di amino acid repeats are contained [46] between PQBP1-WWD and PQBP1-CTD. Human mutations that cause intellectual disability are concentrated at the repeat sequences, as described later in this paper, presumably because repeat sequences in the genome may cause mutations during replication or recombination. Most mutations lead to a deficiency of PQBP-CTD or a decrease in PQBP1 protein level due to non-sense RNA decay [47]. Therefore, the links between the roles of PQBP1-WWD and PQBP1-CTD and between the roles of PQBP1-CTD and the whole PQBP1 molecule would be essential for cellular functions related to intellectual ability and brain size.

## 3. PQBP1 Is an Intrinsically Disordered Protein with a Low Complexity Domain

As described above, PQBP1 is an intrinsically disordered protein (IDP) that does not compose a rigid tertiary protein structure [38,41], but forms intracellular foci similar to nuclear bodies [25] and stress granules [44]. PQBP1 alone can form nuclear foci sequestered from nucleoplasm; however, it can also co-assemble with ataxin 1 (Atxn1) into a nuclear granule, in which the two proteins are separated in a lamellar structure by liquid–liquid phase separation (LLPS) [25,40].

Recently, the involvement of IDPs, such as FUS, hnRNPA1, and TDP43, in the pathology of neurodegenerative diseases has attracted considerable attention [48,49,50]. LLPS of disease causative proteins–which occurs because of the physical characteristics of low complexity (LC) sequences with a high content of specific amino acids–has been proposed as an alternative mechanism for aggregation instead of the classical fibril formation [51]. LLPS may also initiate Tau aggregation [52]. The concept of LLPS enables so-called “aggregates” to behave more dynamically between the soluble state and the assembly state. In this regard, PQBP1 is a pioneering molecule of neurodegeneration-related IDPs that is challenging the paradigms of the aggregation hypothesis [25,40].

## 4. Expression Profile of PQBP1 during Development and in Adulthood

In adult mice, PQBP1 is expressed throughout various organs [46]. In the brain, the expression level is high in regions with dense neuronal populations, such as the hippocampus and cerebellar granular layer, and in situ hybridization has revealed that among the various cell types in the brain, mRNA expression is highest in neurons [46]. In a single cell, the PQBP1 protein is predominantly located in the nucleus of cells; however, it can be translocated to the cytoplasm [46]. Interestingly, PQBP1 is concentrated in RNA granules and translocated to stress granules under certain cellular stresses [53].

During development, PQBP1 is expressed throughout various organs but most highly in the central nervous system [54]. In the developing brain, the highest expression at the mRNA level has been detected in the ventricular and subventricular zone (VZ/SVZ), where neural stem progenitor cells exist [54]. Previously, we further confirmed that among the various central nervous system regions, the highest expression at the protein level is in the VZ/SVZ [55,56]. Furthermore, the PQBP1 expression level is high in developmental bone marrow [54]. These results are consistent with the role of PQBP1 in stem cell proliferation [56], synapse regulation [23], and bone development [57], which are relevant to human symptoms described in the following section.

## 5. Human Gene Mutations and Intellectual Disability

In 2003, the European X-linked MR consortium led by Profs. Kalscheuer and Ropers discovered a genetic linkage between PQBP1 gene mutations at Xp11.23 and the onset of intellectual disability in patients from five of 29 families with syndromic and non-syndromic forms of X-linked mental retardation [47]. The families carrying PQBP1 mutations were clinically similar to Renpenning syndrome [58]. Subsequently, a number of other research groups confirmed the genetic linkage between PQBP1 and intellectual disability [59,60,61,62,63,64,65,66]. This included other intellectual disability syndromes, such as Golabi-Ito-Hall syndrome [64], Hamel syndrome, Proteus syndrome, and Sutherland-Haan syndrome, which led to the concept of the Renpenning syndrome spectrum [61,67]. Mutations of PQBP1 are now categorized into two types (Figure 1). The first type of mutations includes deletion or duplication of dinucleotide repeats or deletion of twenty nucleotides in the hepta-amino-acid repeat regions, which result in a reading frame shift and non-sense RNA decay [47]. The second type of mutation detected in Golabi-Ito-Hall syndrome is the substitution of tyrosine 65–a critical residue of WWD for protein interaction [59]–to a cysteine (Tyr65Cys), which results in structural change, losing the WWD-mediated protein interaction [68,69,70].

Renpenning syndrome spectrum diseases share intellectual disability, microcephaly, lean body, short stature, and small testes as major symptoms (Figure 2). The intellectual disability in patients with these diseases ranges from mild to severe. The microcephaly of Renpenning syndrome spectrum is primary microcephaly without architectural changes in the cerebral cortex [56]. A rare case of PQBP1-linked microcephaly with periventricular heterotopia has been reported, although the patient’s brother did not show periventricular heterotopia [62]. Therefore, PQBP1-linked microcephaly is associated with cortical thickness expansion rather than cortical surface expansion.

The frequency of all PQBP1 gene mutations has not yet been determined, although Renpenning syndrome spectrum is a rare disease (personal communication). However, one report has suggested that the Renpenning syndrome spectrum occurs at a relatively high frequency, near that of Rett syndrome [71] (Figure 2).

Flynn et al. reported duplication of the PQBP1 gene and a phenotype-like Renpenning’s syndrome in a patient [72]. SNP microarray analysis (500 K) of the patient showed a 4.7 Mb duplication at Xp11.22–p11.23, including the PQBP1 gene, which was further confirmed to be duplicated using Multiplex Ligation-dependent Probe Amplification (MLPA) [67]. The patient was diagnosed with intellectual disability and exhibited myoclonic seizures and hyperactivity at two years of age, but normal social behavior at four years of age and moderate intellectual disability (IQ: 35–47) at 11 years of age [72]. Although the patient did not have microcephaly, CT revealed mild cerebral atrophy, and EEG showed a pattern of generalized seizure activity [72]. Another group reported a family with duplication of Xp11.22–p11.23, which includes FTSJ1 and PQBP1 [73]. The family comprised male and female patients with moderate intellectual disability and developmental speech delay [73]. Although not confirmed in these patients’ brains, the possible overexpression of PQBP1 proteins in such patients may be relevant to the phenotypes observed in the Drosophila transgenic model [74].

## 6. Animal Models and Molecular/Cellular/Biological Functions of PQBP1

Various animal models have been developed up to now. In transgenic mice with human PQBP1 driven by ubiquitous gene expression regulatory elements, PQBP1 overexpression exhibited a delayed and slow progressive motor neuron disease-like phenotype [75], while the phenotype is observed in a restrictive number of transgenic mice and may be difficult to be generalized. Subsequent microarray analysis of the motor neuron degeneration process in these mice revealed that mitochondrial abnormalities might occur in neurons [76].

PQBP1 knockdown (KD) mice have also been developed [77], in which a transgene expressing a 498 bp double-stranded RNA produces, when cleaved, endogenously multiple siRNAs that suppress PQBP1. This selectively reduces the level of PQBP1 protein to nearly 50% of that in control mice [77]. The KD mice exhibit abnormal anxiety-related behaviors during the light/dark search, and open-field tests and significant reductions in anxiety-related cognition in the repeated elevated cruciform maze and novel object recognition tests [77]. However, the KD mice did not reveal obvious abnormalities of endoderm-derived organs [77].

Although a generation of PQBP1 knockout mice has not been developed, three conditional knockout mouse models of PQBP1 have been reported [23,24,56]. PQBP1-cKO mice generated using nestin-Cre have microcephaly and exhibit cognitive dysfunctions [56]. Detailed analyses have revealed that the total cell cycle time of neural stem progenitor cells (NSPCs) is elongated, and the frequency of cell division is decreased before delivery [77]. This mechanism is distinct from previously established mechanisms of microcephaly, such as depletion of neural stem cells by enhanced differentiation to neurons, increased cell death of NSPCs and/or neurons, and impaired migration of differentiated neurons, all of which do not exist in Pqbp1-cKO mice [77] (Figure 3). Comprehensive analyses of Pqbp1-cKO mice have revealed that the expression profiles of cell cycle-relevant genes and synapse-relevant genes are widely affected due to loss of PQBP1 [77]. Surprisingly, Pqbp1-cKO mice generated by nestin-Cre exhibit a small body size. Bone CT showed a reduction in bone mass, and bone histology revealed impaired bone formation and deficiency of chondrocytes in Pqbp1-cKO mice [57]. The bone phenotype might be derived from an abnormality of mesenchymal stem cells, a part of which expresses nestin [57].

Synapsin 1-Cre Pqbp1-cKO mice exhibit cognitive dysfunctions, but not microcephaly [23]. This phenotype is quite similar to that of AD model mice, except for Aβ aggregation and neuronal cell death [23]. This similarity will be discussed later in this review, with reference to AD pathology. CX3CR1-Cre Pqbp1-cKO mice were generated to deplete PQBP1 in innate immune cells [24]. Investigation of the phenotypes of CX3CR1-Cre Pqbp1-cKO mice has been focused on the change of response of microglia to Tau [24], which will be described later.

Furthermore, the suppression of a Drosophila homolog of PQBP1 (dPQBP1) by the insertion of piggyBac has produced a Drosophila model [78]. Interestingly, this Drosophila model revealed impaired learning acquisition in olfactory conditioning test, while the short-term, medium-term, long-term, and anesthesia-tolerant memories were not impaired [78]. The underlying mechanism was identified to be reduced expression of the NMDA receptor subunit 1 in projection neurons [78]. These results discovered a new type of cognitive disturbance in which dPQBP1 regulates learning acquisition in projection neurons [78].

In nematodes, two PQBP1 homologs are dominantly located in intestinal fat storage cells [79], and one of the mutants showed a reduction in lipid storage and changes in lipid species [79]. These findings suggest that the nematode model is suitable for lean body analysis of human PQBP1 mutants.

The Pqbp1-cKO mouse models are highly useful for analyzing the pathology and are actually very informative. Their phenotypes are also highly similar to those of human patients. However, if we look at historical development of mouse models for other neurological diseases, humanized knock-in mouse models and primate models might be generated in the future. In this sense, it is of note that there is no humanized knock-in mouse model so far, nor is there a mouse model for a point mutation mimicking Golabi-Ito-Hall syndrome.

## 7. Acquired Reduction in PQBP1 Contributes to Cognitive Abnormalities in AD

Atxn1, whose mutation in the CAG repeat causes SCA1, has been shown to interact with PQBP1 and inhibit its role in transcriptional regulation [25]. PQBP1 interacts with Htt, the causative gene product of Huntington’s disease [12,46], which suggests that these polyQ proteins induce abnormal changes in mRNA transcription and splicing related to synapse functions, similarly to the PQBP1 deficiency in cKO mice.

A homologous pathology was recently revealed in AD by comprehensive phosphoproteome analysis at the ultra-early phase of AD [80]. Extracellular amyloid plaques, which are a pathological feature of AD in model mice, are observed gradually from the age of three months, and cognitive dysfunction becomes apparent at the age of six months. Extracellular amyloid-beta and symptoms are not observed at one month of age. Comprehensive phosphoproteome analysis has revealed that several proteins are already abnormally phosphorylated at the age of one month [23,81].

One such protein is Serine/Arginine Repetitive Matrix 2 (SRRM2), a scaffolding protein for multiple splicing factors, in which the phosphorylation of Ser1068 is abnormally increased in the ultra-early stage of AD [23]. SRRM2 is a protein normally localized to the nucleus; however, this abnormal phosphorylation results shifts its localization to the cytoplasm by blocking the interaction between SRRM2 and the T-complex protein subunit α, which is essential for nuclear translocation [23]. Moreover, the deficiency of SRRM2 destabilizes and reduces PQBP1 in the nucleus of neurons [23]. Eventually, the series of changes directly and severely affects the transcription and splicing patterns of synapse-related genes, as was shown in a PQBP1-conditional knockout model. Furthermore, both PQBP1 and SRRM2 are downregulated in cortical neurons of both human AD patients and AD mouse models. Consistently, treatment of these two AD mouse models (i.e., 5xFAD and APP-KI mice) with AAV-PQBP1 restored RNA transcription and splicing, the synaptic phenotype, and cognitive impairment. These results support the possibility that the acquired reductions in the PQBP1 level are the direct cause of cognitive decline in dementia [23].

## 8. Relationship between Immune Response and PQBP1

Recently, PQBP1 has received attention for its importance as a new immunoregulatory factor [24,82]. Dendritic cells (DCs) that produce immune responses to viral infection are triggered by cyclic GAMP synthase (cGAS) activity-dependent responses when infected with HIV-1. PQBP1 acts as an intracellular receptor binding directly to reverse-transcribed HIV-1 cDNA and initiates innate immune responses [82] (Figure 4). In fact, primary human monocyte-derived DCs from patients with Renpenning syndrome have also been shown to exhibit significantly reduced innate immune responses to HIV-1 [82].

Most notably, a molecular mechanism similar to the one mentioned above was observed in microglia, the innate immune cell in the central nervous system [24]. The Tau protein, which is known to be involved in the pathogenesis of various neurodegenerative diseases including AD and tauopathy, was found to be recognized by PQBP1 in brain microglia and to induce intracellular signaling of the cGAS-STING pathway to induce expression of cytokine genes such as TNF, IL-6 and type 1 IFN for brain inflammation [24] (Figure 4). In mice, the suppression of PQBP1 restricted to microglia in the brains of CX3CR1-Cre cKO mice was shown to suppress Tau protein-induced brain inflammation [24]. This demonstrates the potential of targeting PQBP1 as a new common therapy for neurodegenerative diseases.

## 9. Future Perspectives

An increasing number of experimental and clinical reports have demonstrated the significance of PQBP1 in the biological processes of neurons, microglia, and other cell types, as well as its impact on various types of human diseases. PQBP1 is a representative IDP that is involved in neurological and immunological diseases, which suggests that further studies into PQBP1 could lead to breakthroughs in our understanding of how IDPs function in human diseases. Despite more than 115 reported studies on PQBP1 during the last 20 years (https://pubmed.ncbi.nlm.nih.gov/?term=pqbp1 (accessed on 20 April 2022)) [22], our knowledge of the functional mechanism of PQBP1 still remains insufficient relative to its importance in a growing number of human diseases.

Databases relevant to PQBP1 offer new research possibilities to investigate the functions of this protein. For instance, an examination of various SNP databases (https://www.hgvs.org/central-mutation-snp-databases (accessed on 20 April 2022)) indicates that the effects of a large number of SNPs in the PQBP1 gene are yet to be investigated (https://www.ncbi.nlm.nih.gov/snp/?term=PQBP1, (accessed on 20 April 2022)). So far, there is no knowledge as to how such SNPs influence intellectual disabilities, neurodegenerative diseases, or intelligence. Moreover, protein–protein interaction databases include interacting partners yet to be investigated (https://thebiogrid.org/115393/summary/homo-sapiens/pqbp1.html (accessed on 20 April 2022)), such as amyloid precursor protein, the causative gene product of AD [83,84,85], and the precursor to the pathological signature of AD (Aβ); bromodomain containing 4, a chromosome-binding protein during mitosis [86,87,88,89,90]; and cleavage and polyadenylation specific factors 6 and 7, a factor regulating 3′ RNA cleavage and polyadenylation processing [91,92,93,94]. Investigations into these potential partner proteins will open new avenues for research into biological and pathological mechanisms of PQBP1. Moreover, it would be very interesting if we can reveal how interacting proteins or RNAs are changed by gene mutations of PQBP1, and the findings will lead to further understanding of molecular basis of intellectual disabilities and other neurological disorders.

## Figures and Tables

**Figure 1 ijms-23-06227-f001:**
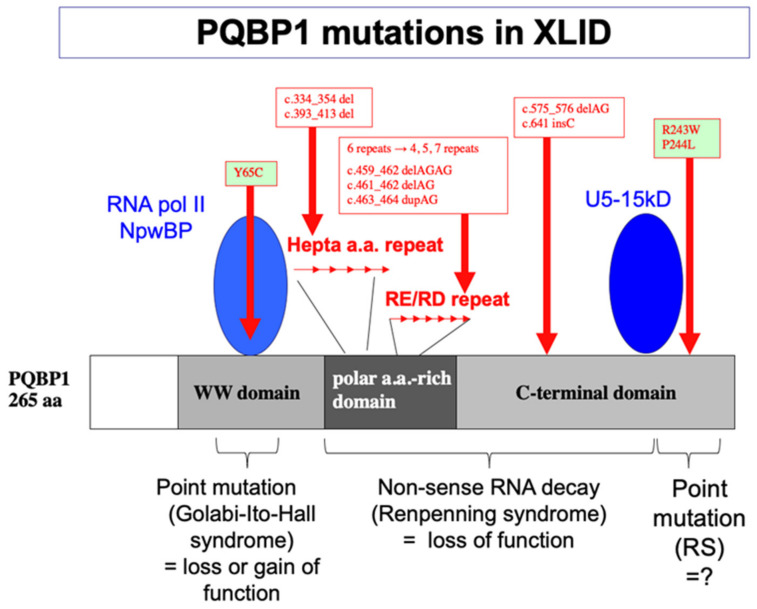
Domains of PQBP1 and human mutation sites. Reported human mutations of PQBP1 gene are summarized in correspondence with domains. A point mutation at Tyrosine 65 critical for the structure of WW domain causes Golabi-Ito-Hall syndrome. Deletions or insertions in the polar amino acid rich domain or in the C-terminal domain cause frame shifts, which lead to the reduction of mRNA due to non-sense RNA decay and cause Renpenning syndrome. Point mutations at the tail of the C-terminal domain also cause Renpenning syndrome, while the direct effect on PQBP1-mRNA and -protein remains unknown.

**Figure 2 ijms-23-06227-f002:**
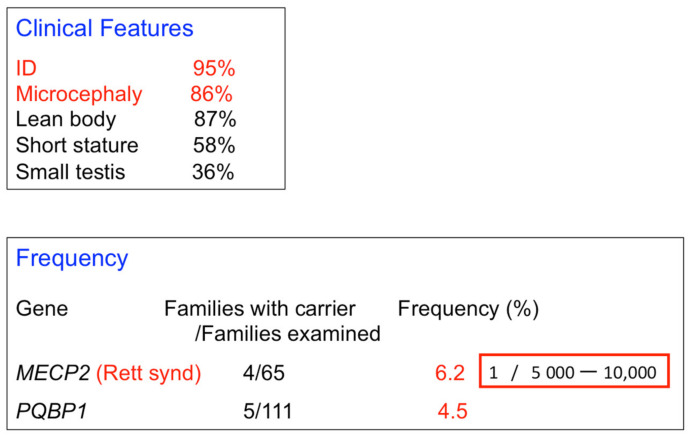
Clinical symptoms and frequency of patients carrying PQBP1 mutations. Clinical features of patients carrying PQBP1 gene mutations are summarized. Intellectual disability (ID) and microcephaly are almost essential, while lean body and short stature are highly frequent. The real incidence of patients carrying PQBP1 gene mutations is not determined. However, a report by de Brouwer and colleagues [66] suggested it to be relatively high.

**Figure 3 ijms-23-06227-f003:**
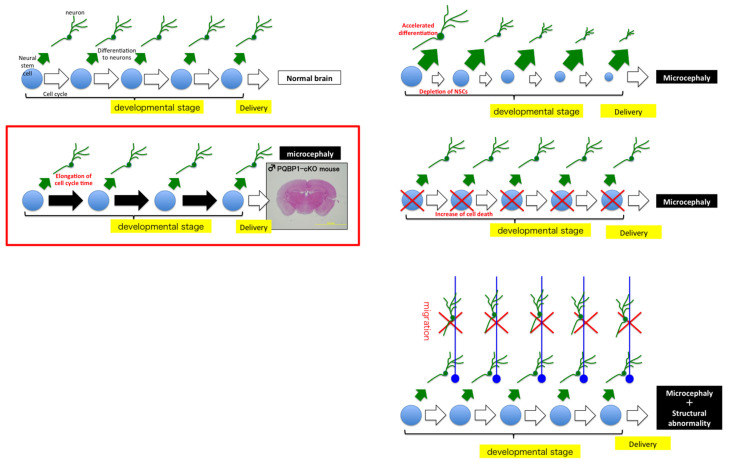
The mechanism of PQBP1-linked microcephaly and the three major mechanisms of microcephaly published previously. (**Upper left**) In normal state, neurons are produced from neural stem cells at the constant differentiation rate. (**Lower left**) The microcephaly mechanism of PQBP1-cKO mice is shown. Due to the elongation of total cell cycle time, even though the differentiation rate is not changed, the total number of neurons produced from neural stem cells is reduced. The right side panels show previously known mechanisms of microcephaly. (**Right upper**) In the case where the differentiation rate to neurons is increased, the neural stem cell pool is depleted earlier than in normal state, and the total number of neurons produced from neural stem cells is reduced. (**Right middle**) When cell death of neural stem cells occurs, the total number of neurons produced from neural stem cells is reduced. (**Right lower**) When migration of differentiated neurons is impaired, the cortical layer structure becomes abnormal, and the abnormality leads to microcephaly.

**Figure 4 ijms-23-06227-f004:**
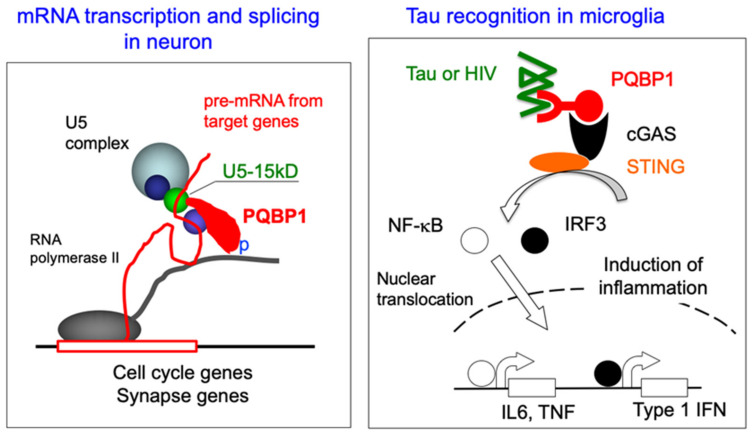
PQBP1 functions in the nucleus of neurons and the cytoplasm of microglia. (**Left**) In neurons, PQBP1 functions as an adaptor between transcription and splicing. Immediately after hnRNA (pre-mRNA), splicing complexes assemble for cleavage of intron. PQBP1 binds to the C-terminal tail of RNA polymerase II in a phosphorylation-dependent manner and promotes the splicing of specific types of hnRNAs. PQBP1 is known to directly interact with U5-15kD in U5 spliceosome. (**Right**) In microglia, PQBP1 is known to play another role as an intracellular receptor for pathogens. PQBP1 recognizes cDNA of HIV to trigger cGAS-STING signaling pathway for inflammation. Recently it is shown that PQBP1 recognizes the Tau protein, which is implicated in Alzheimer’s disease and various Tauopathies in a similar manner than if the Tau were a pathogen. It is yet to be investigated whether PQBP1 triggers cGAS-STING pathway in neurons or whether PQBP1 regulates splicing in microglia.

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
