# Peer review of "PQBP1: The Key to Intellectual Disability, Neurodegenerative Diseases, and Innate Immunity"

_ijms, 2022, doi:10.3390/ijms23116227_

Round 1

Reviewer 1 Report

This is a comprehensive, balanced and very pleasantly written review on the normal function and known disease involvement of the PQBP1 protein. It provides an up-to-date perspective on this protein and its multiple links to nervous system disorders and, more recently established, immune response to infection and toxic stimuli. In addition, the review provides a good overview of the standing questions in the field and future perspectives.

I have only a few suggestions of topics that the authors could briefly address to provide an even more balanced discussion of the topic, and reinforce the translational implications of the  findings:

  1. How are PQBP1's protein-protein interactions or protein-RNA interactions affected by the mutations that cause Renpenning syndrome and related phenotypes in humans? Is there information on what happens to the protein partners in human patient or animal models' brains?
  2. Do SNP's in the PQBP1 gene act as modifiers of human polyglutamine diseases?
  3. How do authors see the validity (concept validity, face validity and predictive validity) of the currently available animal models for the study of Renpenning syndrome, namely for our understanding of how human mutations cause the disease? Does the community need to invest in developing further models or are these sufficient, for instance for pre-clinical drug discovery studies?
  4. Modulation of the expression of PQBP1 is suggested to be a therapeutic target for human neurodegenerative diseases. However, the effects of this modulation seem to go in opposite ways for two aspects known to be relevant for Alzheimer's disease: the microglial activation induced by tau (where suppression of PQBP1 expression seems to be beneficial) and the neuronal response to Abeta exposure (where overexpression of PQBP1 seems to be protective). How do authors reconcile these observations? How do they see the design of therapeutic approaches targeting PQBP1? Is this a druggable target or would gene therapy be the approach to use?
  5.  

Author Response

Reviewer 1

Comments and Suggestions for Authors

This is a comprehensive, balanced and very pleasantly written review on the normal function and known disease involvement of the PQBP1 protein. It provides an up-to-date perspective on this protein and its multiple links to nervous system disorder sand, more recently established, immune response to infection and toxic stimuli. In addition, the review provides a good overview of the standing questions in the field and future perspectives.

I have only a few suggestions of topics that the authors could briefly address to provide an even more balanced discussion of the topic, and reinforce the translational implications of the findings:

>>> Thank you very much for the kind evaluation.

  1. How are PQBP1's protein-protein interactions or protein-RNA interactions affected by the mutations that cause Renpenning syndrome and related phenotypes in humans? Is there information on what happens to the protein partners in human patient or animal models' brains?

>>> We appreciate very much this thoughtful comment. Unfortunately, we and other groups have not performed comprehensive survey for changes of interacting partners by ID-linked gene mutation. This is a very interesting suggestion for possible experiments in the future, so we borrowed the idea (though I also have already submitted the similar idea to a grant in Japan) and described it in the section of future perspective.

  1. Do SNP's in the PQBP1 gene act as modifiers of human polyglutamine diseases?

>>> This is again very interesting suggestion, so we included a sentence for this idea.

  •  
  1. How do authors see the validity (concept validity, face validity and predictive validity) of the currently available animal models for the study of Renpenning syndrome, namely for our understanding of how human mutations cause the disease? Does the community need to invest in developing further models or are these sufficient, for instance for pre-clinical drug discovery studies?

>>> The conditional knockout models are highly useful and informative for analyzing the pathology. The phenotypes are highly similar to those of human patients. However if we look at historical development of mouse models for other neurological diseases, humanized knock-in mouse models and primate models might be generated in the future. We comment on this issue at the end of Section 5.

  •  
  1. Modulation of the expression of PQBP1 is suggested to be a therapeutic target for human neurodegenerative diseases. However, the effects of this modulation seem to go in opposite ways for two aspects known to be relevant for Alzheimer's disease: the microglial activation induced by tau (where suppression of PQBP1expression seems to be beneficial) and the neuronal response to Abeta exposure(where overexpression of PQBP1 seems to be protective). How do authors reconcile these observations? How do they see the design of therapeutic approaches targeting PQBP1? Is this a druggable target or would gene therapy be the approach to use?

>>> We thank the deep interest of the reviewer about PQBP1 for therapeutic use. We agree with the comment. Actually we described the result of AAV-PQBP1 in our paper (Jin et al, Nature Commun 2021) that our AAV-PQBP1 does increase PQBP1 in neurons but does not increase PQBP1 in microglia. This is lucky, while we recognize that we need to further sophisticate the method for selective expression such as using cell-type specific enhancer-promoter with a safety system like Tet-off. We did not include this discussion in the paper due to restriction of intellectual property.

Reviewer 2 Report

This review is well written and gives a comprehensive summary of recent findings of PQBP1. I suggest minor changes before acceptance for publication.

Comments:

Page 4, 5th line started with the sentence “These results are consistent with…”: please add the corresponding references.

Figure 1: since the diagram does not show the “structure” of PQBP1 but the domain information, I suggest to change the title and add figure legend to give a summative explanation of this diagram.

Figure 2: similar as the comment above, it is better to move the references to the additional figure legend. And this comment applies to all the figures that figure legend is strongly suggested to help readers to understand the figures.

Author Response

Reviewer 2

Comments and Suggestions for Authors

This review is well written and gives a comprehensive summary of recent findings ofPQBP1. I suggest minor changes before acceptance for publication.

>>> Thank you very much for the kind evaluation.

Comments:

Page 4, 5 line started with the sentence “These results are consistent with…”:please add the corresponding references.

>>> We added references.

Figure 1: since the diagram does not show the “structure” of PQBP1 but the domain information, I suggest to change the title and add figure legend to give a summative explanation of this diagram.

>>> We changed the title and added the figure legend.

Figure 2: similar as the comment above, it is better to move the references to the additional figure legend. And this comment applies to all the figures that figure legend is strongly suggested to help readers to understand the figures.

>>> We removed references from Figure 2, and added legends to Figure 2-4

Reviewer 3 Report

H.Tanaka and H. Okazawa have reviewed the role of PQBP1 in Intellectual Disability, Neurodegenerative Diseases, and Innate Immunity. Overall, the manuscript is well written, and it provides the recent update on the role of PQBP1 ranging from intrinsic disorder protein to intracellular receptor. However, there are several questions to be cleared before the final acceptance of the manuscript.

Major comments:

11.  Resolution of Figure 3 must be improved by redrawing. Also, sub sectioning of the Figure 3 must be mentioned and marked accordingly in the description of “6. Animal models and molecular/cellular/biological functions of PBP1”. Currently it is not clear to follow.

  2. Cite the reference for this below sentence in the introduction

In addition, PQBP1 has been implicated in neurodegenerative diseases, including Alzheimer’s disease (AD), tauopathy, and polyQ diseases, as a common mediator across multiple disease pathologies

33. Cite the reference for the first idea and second the idea in the introduction. Currently there is no reference for further understanding

The first idea was “sequestration,” whereby normal proteins are co-segregated

into the inclusion body of polyQ proteins and lose their physiological functions.

The second idea was that mutant polyQ proteins acquire higher (or lower) affinity for their binding partner molecules, and such gains in abnormal interactions (or loss of physiological

interactions) lead to gains in toxicity.

44.  In the section 6. Animal models and molecular/cellular/biological functions of PBP1,

“PQBP1 not PBP1”

55. Better explanation for the figure 4 in the description is needed. Currently the relevance for figure on the right side of figure 4 (mRNA transcription and splicing in neuron) in the description is not convincing.

66. Remove the word “top journal” in the below sentence. Journals are journals.

“Despite more than 115 reported studies on PQBP1, including those published in top journals during the last 20 years”

Minor comments

7.       Drosophila sometimes in italics and sometimes it is not. Check for uniformity

8.       Full name for SRRM2

9.       Check the phrase for English: that the total cell cycle time of neural stem progenitor cells

(NSPCs) is “elongated”

Abbreviated form of ID and IDP is confusing. I recommend ID full name Intellectual disability and abbreviated form for intrinsic disorder protein (IDP).

1In Figure 2, keep the uniformity in the citation sometimes it is et al. sometimes it is et al,

1 In the section 4. Expression profile of PQBP1 during development and in adulthood,

“…. synapse regulation, and bone development, which are relevant to human symptoms”. human symptom of which disease?

1In the section 3. PQBP1 Is an intrinsically disordered protein with a low complexity domain-“change the capital letter in Is”.

1Check the phrase for English: PQBP1 is expressed throughout various organs but “most

Highly”

1 In the section 2. Molecular structure and cellular function of PQBP1, describe in parenthesis what is B*?.

1In the section 2. Molecular structure and cellular function of PQBP1,

“……the whole PQBP1 molecule would be essential for cellular functions related functions related to intellectual ability and brain size”.

Are there any literature evidence to support brain size change?  Is it certain region of the brain or the entire brain?

1 Describe the full name of abbreviation in the first mentioning. For example, IDP in the section 2. Molecular structure and cellular function of PQBP1.  Also, I suggest to full name of abbreviation CD, NMR and EOM. While CD and NMR is widely used EOM is relatively unknown.

Author Response

Reviewer 3

H.Tanaka and H. Okazawa have reviewed the role of PQBP1 in Intellectual Disability, Neurodegenerative Diseases, and Innate Immunity. Overall, the manuscript is well written, and it provides the recent update on the role of PQBP1 ranging from intrinsic disorder protein to intracellular receptor. However, there are several questions to be cleared before the final acceptance of the manuscript.

>>> Thank you very much for the kind evaluation.

Major comments:

  1. Resolution of Figure 3 must be improved by redrawing. Also, subsectioning of the Figure 3 must be mentioned and marked accordingly in the description of “6. Animal models and molecular/cellular/biological functions of PBP1”. Currently it is not clear to follow.

>>> Quality of Figure 3 was improved.

>>> Following suggestion from the other reviewer, we added figure legend for Fig 3, in which we described the details and probably aid the understanding of mechanisms.

>>> In the previous version, we have already mentioned and marked Figure 3. I have added the comment at the position.

  1. Cite the reference for this below sentence in the introduction

In addition, PQBP1 has been implicated in neurodegenerative diseases, including Alzheimer’s disease (AD), tauopathy, and polyQ diseases, as a common mediator across multiple disease pathologies

>>> We added references.

  1. Cite the reference for the first idea and second the idea in the introduction. Currently there is no reference for further understanding

The first idea was “sequestration,” whereby normal proteins are co-segregated into the inclusion body of polyQ proteins and lose their physiological functions.The second idea was that mutant polyQ proteins acquire higher (or lower) affinity for their binding partner molecules, and such gains in abnormal interactions (or loss of physiological interactions) lead to gains in toxicity.

>>> We added references.

  1. In the section 6. Animal models and molecular/cellular/biological functions of PBP1,“PQBP1 not PBP1”

>>> Thank you for pointing out the error. We corrected it.

  1. Better explanation for the figure 4 in the description is needed. Currently the relevance for figure on the right side of figure 4 (mRNA transcription and splicing in neuron) in the description is not convincing.

>>> We described the details in figure legends.

  1. Remove the word “top journal” in the below sentence. Journals are journals. “Despite more than 115 reported studies on PQBP1, including those published in top journals during the last 20 years”

>>> We followed the comment.

Minor comments

  1. Drosophila sometimes in italics and sometimes it is not. Check for uniformity

>>> Drosophila became non-italic in all positions.

  1. Full name for SRRM2

>>> Full name was added.

  1. Check the phrase for English: that the total cell cycle time of neural stem progenitor cells (NSPCs) is “elongated”

>>> Several native editors have checked.

10   Abbreviated form of ID and IDP is confusing. I recommend ID full name Intellectual disability and abbreviated form for intrinsic disorder protein (IDP).

>>> We followed the advice.

11 In Figure 2, keep the uniformity in the citation sometimes it is et al. sometimes it is et al,

>>> Following the comment of another reviewer, the references were depleted from Figure 2.

12  In the section 4. Expression profile of PQBP1 during development and in adulthood, “…. synapse regulation, and bone development, which are relevant to human symptoms”. human symptom of which disease?

>>> We described it.

13 In the section 3. PQBP1 Is an intrinsically disordered protein with a low complexity domain-“change the capital letter in Is”.

>>> We changed it.

14 Check the phrase for English: PQBP1 is expressed throughout various organs but “most Highly”

>>> This phrase “most highly” was recommended by native editors.

15  In the section 2. Molecular structure and cellular function of PQBP1, describe in parenthesis what is B*?.

>>> B* complex is a common sense to RNA splicing field researchers. The spliceosome complex is shifting sequentially during the process with shifting components, and there is the B* complex.

16 In the section 2. Molecular structure and cellular function of PQBP1,“……the whole PQBP1 molecule would be essential for cellular functions related functions related to intellectual ability and brain size”. Are there any literature evidence to support brain size change?  Is it certain region of the brain or the entire brain?

>>> As described in the text already, “The microcephaly of Renpenning syndrome spectrum is primary microcephaly without architectural changes in the cerebral cortex 52.” The microcephaly is not regional but whole brain. Regarding the whole molecular function, the depletion of C-terminal domain in the case of frame shift mutations (previous ref 43 → new ref 47) support the idea.

17  Describe the full name of abbreviation in the first mentioning. For example, IDP in the section 2. Molecular structure and cellular function of PQBP1.  Also, I suggest to full name of abbreviation CD, NMR and EOM. While CD and NMR is widely used EOM is relatively unknown.

>>> We added the full names.
